# Smart Active Vibration Control System of a Rotary Structure Using Piezoelectric Materials

**DOI:** 10.3390/s22155691

**Published:** 2022-07-29

**Authors:** Ali Hashemi, Jinwoo Jang, Shahrokh Hosseini-Hashemi

**Affiliations:** 1Department of Civil, Environmental and Geomatics Engineering, Florida Atlantic University, Boca Raton, FL 33431, USA; ahashemi2020@fau.edu; 2Department of Mechanical Engineering, Iran University of Science and Technology, Tehran 13114-16846, Iran; shh@iust.ac.ir

**Keywords:** active vibration control, smart structure, analytical vibration analysis, transfer function method, piezoelectric sensor-actuator

## Abstract

A smart active vibration control (AVC) system containing piezoelectric (PZT) actuators, jointly with a linear quadratic regulator (LQR) controller, is proposed in this article to control transverse deflections of a wind turbine (WT) blade. In order to apply controlling rules to the WT blade, a state-of-the-art semi-analytical solution is developed to obtain WT blade lateral displacement under external loadings. The proposed method maps the WT blade to a Euler–Bernoulli beam under the same conditions to find the blade’s vibration and dynamic responses by solving analytical vibration solutions of the Euler–Bernoulli beam. The governing equations of the beam with PZT patches are derived by integrating the PZT transducer vibration equations into the vibration equations of the Euler–Bernoulli beam structure. A finite element model of the WT blade with PZT patches is developed. Next, a unique transfer function matrix is derived by exciting the structures and achieving responses. The beam structure is projected to the blade using the transfer function matrix. The results obtained from the mapping method are compared with the counter of the blade’s finite element model. A satisfying agreement is observed between the results. The results showed that the method’s accuracy decreased as the sensors’ distance from the base of the wind turbine increased. In the designing process of the LQR controller, various weighting factors are used to tune control actions of the AVC system. LQR optimal control gain is obtained by using the state-feedback control law. The PZT actuators are located at the same distance from each other an this effort to prevent neutralizing their actuating effects. The LQR shows significant performance by diminishing the weights on the control input in the cost function. The obtained results indicate that the proposed smart control system efficiently suppresses the vibration peaks along the WT blade and the maximum flap-wise displacement belonging to the tip of the structure is successfully controlled.

## 1. Introduction

The control of unwanted vibrations of wind turbine (WT) blades is crucial in ensuring highly efficient and cost-effective WT, with increased structure lifetime. Blade’s vibrations can cause extreme operation instability of WT and even induce catastrophic failure of the whole turbine, which must be prevented. Many studies have used passive, active, and semi-active control systems to control the vibrations of sensitive rotary structures [1,2,3,4,5,6,7,8]. Further, to damp the undesired vibrations of structures, proper actuators are needed to apply controlling forces. Due to changing blade dynamics and excitation conditions, passive control methods have had limited use for rotating structures. However, active control methods typically provide a large vibration reduction, which can be of great help for damping unwanted vibrations of sensitive structures. This control process must use smart materials. Among different smart materials, piezoelectrics (PZTs) have three unique advantages: lightweight, low cost, and convenient usage. They provide sensing and actuating features, which make them appropriate materials for vibration control. PZT transducers have been widely used as sensors and actuators in active vibration control (AVC) systems owing to their special properties [9,10,11,12,13]. PZT transducers have been used in various shapes and forms, such as perfectly bounded layers along surfaces of structures or patches in different sizes. PZT-based vibration alleviation can reduce the unwanted vibration levels of rotating structures, thereby reducing the risk of high-cycle fatigue. In this study, we propose an AVC system with PZT patches as sensors and actuators. To simulate implementation of active control rules on structures, we propose an innovative semi-analytical method for projecting the actual shape of a WT blade to the same scale of a Euler–Bernoulli beam. The proposed method is employed to derive an analytical solution for the WT blade’s dynamic response and then to apply controlling rules to the WT blade. The implementation of controlling rule on the WT blade is not feasible when finite element (FE) commercial software, such as ANSYS, is used due to high computational cost. Hence, the proposed transfer function method is employed as a surrogate to obtain the WT blade’s controlled deflections in this study. Therefore, in the first step, we focus on the development of a new semi-analytical solution to obtain WT blade’s lateral deflection under different external loadings. Second, a suitable control system is designed to suppress the WT blade’s unwanted transverse deflection.

The derivation of analytical solutions for vibration and elasticity analysis of structures have always been considered an important field of research in structure analysis and design. Analytical solutions of dynamic responses in various structures have attracted considerable attention for saving computational cost, especially for large-scale structures. However, the derivation of analytical solution for many structures could be very challenging owing to their intricate geometry. Moreover, in recent years, renewable energy, such as wind energy, has been widely considered as one of the alternative energy resources due to a dramatic increase in global warming. For this reason, various types of WT, such as off-shore and on-shore WT with different sizes and power generations, have been investigated in recent years. Bigger, complex WTs have often gained more popularity due to being more efficient in power generation. Large-scale WTs can suffer from significant vibrational deflections, particularly on their blade’s edges. These unwanted vibrations can cause severe structural damage and failure of the power generation systems. Due to their complex shapes and continuous interaction between wind flows and their blades, analysis of WT blade’s dynamic and vibrational behaviors are tremendously difficult. A new understanding of analytical solutions for analyzing the dynamic behaviors of these complex blade shapes can contribute to the analytical analysis of different mechanical and structural systems: twisters and heavy solid structures. Compared with analytical methods, the vibration of these twister structures has primarily been investigated using FM simulation and other numeric methods, such as finite difference (FD) and differential quadrate. To control the structures’ vibrations, three controlling systems, including passive, active, and semi-active, have been considered. These controlling systems use various control rules, such as proportional pulse derivative (PD), fuzzy logic control, and sliding mode control. The active control method outperforms other methods for WT blades due to various environmental excitation conditions and the provision of large vibration reduction [14].

At the first step, we propose a unique semi-analytical solution for WT blades, which can describe lateral blade movements under specific external forces. In general, a semi-analytical solution technique is proposed to solve linear partial differential equations. The semi-analytical method depends on analytically solving the equations derived by discretizing the spatial coordinates of partial differential equations. Semi-analytical approaches have a significant advantage over numerical methods regarding solution time. This study demonstrates how to project a WT blade’s real form to the same size of a Euler–Bernoulli beam to derive the WT blade’s dynamic and vibration responses. The blade of the GE 1.5-megawatt model [15], which has 45-m blades on a 9-m tower, is considered in this study. The blade’s material is assumed to be steel to simplify the equations’ derivation. Plumbum–Titania–Zirconia is considered as PZT sensor–actuator patches to sense and actuate structures. At the second step, to alleviate vibrations and transverse deflection of the WT blade with PZT patches, a linear quadratic regulator (LQR) control method is developed in this study. Due to decreasing computational cost, the proposed semi-analytical method is used to implement control rules and obtain controlled transverse vibration of the WT blade’s actual shape instead of using the blade’s FE model.

## 2. Background

Rotating structures, which have significance in many practical applications such as turbine blade, airplane propellers, and robot manipulators, have been investigated for a long time. The vibrations of twister beams have been widely studied with different types of beam models, such as Euler–Bernoulli beams, using analytical, semi-analytical solutions and numerical methods. Most studies considered the Euler–Bernoulli and Timoshenko models to describe twister and underloading beams without considering shear deformations. The natural frequencies, mode shapes, and maximum vertical displacements of rotary beams have all been studied to better understand their dynamic behaviors. However, obtaining the dynamic behaviors of rotary beams is difficult due to various environmental factors, including different wind flows and gravity loads. Huang et al. [16] obtained the natural frequencies of a Euler–Bernoulli beam during high-speed rotation using an exponential series solution. A static analysis based on the piezoelectricity and elasticity was conducted by Her and Lin [17] to evaluate the loads induced by PZT actuators to simply supported laminate rectangular plate. An analytical solution of the vibration response of the hosted structure under time harmonic electrical loading is achieved and compared with the FE analysis results to validate the current approach. Arvin [18] studied the nonlinear free vibrations of a rotating beam. He performed the von Kamran–strain displacement relations and derived nonlinear motion equations using Hamilton’s principle. Da Silva [19] presented a systematic and versatile research of a helicopter rotor blade’s responses. First, he presented complete nonlinear partial differential equations governing the blade’s motion, which considers the geometric nonlinearities occurring due to deformation, and the system’s equilibrium solution was then specified by the system. Yigit et al. [20] studied the flexural movement of a radially rotating beam connected to a rigid body. By using the extended Hamilton principle, fully coupled nonlinear motion equations were obtained. Hanagud [21], Baruh [22], and Choura et al. [23] studied dynamic models of rotating Euler–Bernoulli beams without considering centripetal forces on the beams. Most studies did not investigate the fluid–structure interaction [24]. Song et al. [24] established an elaborate model in understanding the fluid–structure interaction between a structure and air flow. Using the open-source OpenFOAM tools coupled with the arbitrary mesh interface framework, Wang et al. [25] provided numerical simulations of WT blade-tower interaction. The vibration analysis of WT blades or other rotary beam shape structures can be divided into two main parts. The first ones includes edgewise vibration that occurs outside the beam’s rotation circle, and the second one comprises flap-wise vibration that occurs in the rotation plate. Lee et al. [26] investigated flap-wise vibration of a composite rotational Euler–Bernoulli beam and the relationship between rotational speed and natural frequencies. Asr et al. [27] suggested prestressing in the blade structure of the H-Darrieus WT for axial compression stress. Their research presented a structural comparison in terms of their dynamic vibrational response among reference and prestressed turbine rotor configurations. Jokar et al. [28] obtained the dynamic modeling and free vibration analysis of horizontal axis WT blades in the flap-wise direction by evaluating blade kinetic and potential energies and by using Hamilton’s principle. Farsadi et al. [29] conducted a semi-analytical solution for the free vibration analysis of uniform and symmetric pre-twisted rotating TW. They used the Green–Lagrange strain tensor to derive the strain field of the system and Hamilton’s principle to derive the governing equations of the dynamic system. Badarinath et al. [30] proposed a surrogate model for the FE analysis of a cantilever beam to define the structure’s dynamic characteristics. Afzali et al. [31] derived a vibration model for an H-rotor/Giromill blade. They assumed that the blade under transverse bending and twisting deformation was treated as a uniform straight elastic Euler–Bernoulli beam. The derivation of the energy equations and simplified aerodynamic models for bending and twisting blades have been distributed, and Lagrangian equations have been extended to assumed modal coordinates to derive nonlinear motion equations for bending and twisting blades [32,33,34,35,36]. Meksi et al. [37] derived the equations of motion of functionally graded sandwich plates from Hamilton’s principle based on a new shear deformation plate theory. A semi-analytical solution is proposed by Wang et al. [38] for the vibration analysis of infinite partially electrode circular AT-cut quartz plates by solving the two-dimensional (2D) scalar differential equation derived by Tiersten and Smythe. Alsabagh et al. [39] implemented the Rayleigh–Ritz method for a typical 5-MW WT blade and developed MATLAB codes. Furthermore, they obtained natural frequencies for both flap- and edgewise vibrational behaviors. Chen et al. [40] derived a dynamic model of curved beams using the absolute nodal coordination formulation based on the radial point interpolation method (RPIM). Chen et al. [41] investigated the free vibration of rotating tapered Timoshenko beams using the technique of variational iteration. Mokhtar et al. [42] investigated the rotor–stator interaction phenomenon in the FE framework using Lagrange multiplier based on contact mechanics. Tang et al. [43] presented a developed approach for the identification of the operational blade vibration modes by measuring the vibrational displacements with a non-contact single-point laser sensor during the wear process. Liu et al. [44] studied structural vibration by establishing a dynamic equilibrium equation of a coupled system comprising a five-leaf blade. A blade excitation force model comprising transverse and vertical excitation forces was created using a quasi-steady method. Zheng et al. [45] explored the Hamilton principle and FE method using a rotating pre-twisted and leaned cantilever beam model (RPICBM) with the flap-wise–chordwise–axial–torsional coupling. They validated the efficacy of the model through comparisons with the literature and the FE models in ANSYS. Warminski et al. [46] studied dynamics of a rotor comprising a flexible beam linked to a slewing rigid hub based on extended Bernoulli–Euler theory for a slender beam model, which considers a nonlinear curvature, synchronized transverse, and longitudinal oscillations, and the hub’s non-constant angular velocity. In different engineering structures, rotating composite beams and blades have numerous applications [47]. Rafiee et al. [47] presented a comprehensive analysis of scientific papers on rotating composite beams, as presented in the past decades. For the flexural study of a sandwich beam combined with a PZT layer, Wang and Quek [48] proposed a fundamental mechanics model. The Maxwell equation was employed in the formulation to extract the PZT potential’s distribution. Shamshirsaz et al. [49] studied piezoelectric-generated Lamb wave in a Timoshenko beam analytically, experimentally, and using finite element simulation. They showed that exciting the structure in high frequency prevents to overlap the emitted waves from the actuator and reflected waves from the boundaries in beam shape structures. Chen et al. [50] developed a semi-analytical solution of the AG–WEC’s dynamic features via the frequency–time domain analysis based on the potential flow principle. Huang et al. [51] presented a high-order FE model and sliding model control method for a rotating flexible structure with the PZT layers to effectively reduce the vibration. Lin [52] used proportional and derivative controls to damp vibration of a rotating beam using a pair of PZT sensor and actuator layers. Bendine et al. [53] investigated the AVC of a composite plate using discrete PZT patches. An FE model with PZT patches was derived based on first-order shear deformation theory, and a damping effect on the composite plate was provided using PZT actuators and LQR control algorithms. Larbi and Deu [54] presented an efficient electromechanical FE formulation for the dynamic analysis of a cantilever beam with PZT patches. Ma et al. [55] investigated an AVC of a moving cantilever beam with PZT ceramics as an actuator using a pulse derivative closed-loop feedback system. Sivrioglu et al. [56] successfully attempted to attenuate a blade’s vibration with a PZT actuator patches using a robust multi-objective control. Sharma et al. [57] measured the controlled response of a fuzzy logic controller for various PZT materials in AVC and compared them with each other. Cui et al. [58] proposed a smart AVC system with a new robust controller in which the system comprised PZT materials, signal conditioning, and an embedded sensor system. Pu et al. [59] explored an adaptive vibration control system for smart structure using a surface-bonded PZT actuator and a filtered-U least mean square algorithm. Khan and Kim [10] studied the AVC of a piezo-bonded laminated composite in the presence of sensor partial debonding and structural delamination. Awada et al. [60] explored the advantages, drawbacks, and challenges of the existing vibration control systems for wind turbines. They present the problems associated with wind turbine vibration in detail and review the different mitigation solutions. Lee [61] studied the vibration control of the stiffened wind blades subjected to a wind load with PZT sensors and actuators to mitigate fluctuations in loading and add damping to the blade. He developed a laminated composite blade with a shear web and the PZT layers embedded on the top and bottom surfaces act as a sensor and actuator. Tong and Zhao [62] designed a blade pitch controller employing linear parameter-varying synthesis techniques for a hydrostatic wind turbine with a barge platform, which was based on the LIDAR (Light Detection and Ranging) preview on the wind speed. An overview of the typical structural characteristics of a modern wind turbine tower and the vibration properties of towers in harsh multi-hazard environments were presented by Malliotakis et al. [63]. The Author conducted a comprehensive review of the most promising passive, active, and semi-active vibration control methods focusing on recent advances around novel concepts and analyses of their performance under multiple environmental loads, including wind, waves, currents, and seismic excitations. An active damper named the twin rotor damper was presented by Bai et al. [64] to reduce vibration in wind turbine towers. The twin-rotor damper was used as a damping system for a single degree of freedom (SDOF) oscillator. Golnary and Tse [65] proposed a method to actively control the lateral vibrations of the tower by using the generator torque and also simultaneously reduce fluctuations in the output power. Liu and Gong [66] studied vibration and control of cantilever blade with bending-twist coupling based on the trailing-edge flap by a restricted control input. Vibration control was investigated based on a linear matrix inequation approach utilizing restricted control input (LMI/RCI). Mu et al. [67] designed an AVC system to improve the stability of a Spar-type floating wind turbine. In this study, the vibration control of WT structure was analyzed by using tuned mass damper active control and the designed active LQR algorithm. An active control system was designed by Brahem et al. [68] based on a full-state LQR controller, which was applied to a rotary beam to alleviate its vibration. Shakir and Saber [69] developed a linear coupled FE model by ANSYS for the PZT actuation of a cantilever beam to study smart beam behavior in open and closed-loop cases. Qiu et al. [70] developed a sliding mode control strategy to damp the vibration of a PZT flexible cantilever plate. The FE modeling was used to simulate the controller on bending and torsional vibration in their research. Fu et al. [71] presented a velocity self-sensing method and experimental verification of a vibration isolation system in which a self-sensing actuator is designed to isolate the vibration with varying frequencies according to the dynamic vibration absorber structure. Ghaderi and Ghatei [72] presented an integrated virtual synchronization/LQR method to identify the system’s unknown physical parameters and vibration control of structures based on estimated parameters.

## 3. Theory and Modeling

To obtain analytical solutions for dynamic and vibration responses to any external forces and excitations, unlike complex structures such as WT blades, the Euler–Bernoulli beam has been widely employed. Since the Euler–Bernoulli beam theory is based on a few key assumptions and the beam has a simple geometric form, analytical study of the Euler–Bernoulli beam is achievable. The Bernoulli–Euler beam theory assumes that “plane sections remain parallel” and that deformed beam angles (slopes) are small; thus, shear deformations may be disregarded. The novelty of this study includes the projection of a WT blade’s deformations and lateral deflections to the same scale of the Euler–Bernoulli beam and then using the proposed mapping system to apply control rules on the WT blade and obtain the blade’s controlled vibration movements. We propose a unique transfer function matrix to undertake the projection of a WT blade to a Euler–Bernoulli beam. Lateral deflections of the Euler–Bernoulli beam are transferred to the WT blade by using the proposed transfer function matrix. This projection enables us to simply obtain the dynamic and vibration responses of a WT blade using the Euler–Bernoulli beam. Next, to damp unwanted lateral deflection of the WT blade, LQR is considered in this study. An AVC is proposed in this study by combining the PZT material as an actuator and a designed LQR control method. The transfer function method is employed to implement the control system on the structure owing to its decreased computational costs.

To develop the transfer function matrix, both the WT blade and the Euler–Bernoulli beam need to be excited with the same external excitation. Then, the lateral movement of selective specific nodes on their surfaces is obtained. In the proposed method, the Euler–Bernoulli beam should have the same length as the WT blade, which is 45 m in this study. We then obtain the lateral deflection of the Euler–Bernoulli beam analytically. FE models are used to calculate the counterpart of the WT blade. PZT patches are used to apply external excitations and obtain the dynamic responses of both the WT blade and Euler–Bernoulli beam. Owing to their electromechanical properties, PZT materials are effective sensors and actuators. PZT sensors and actuators are patched on the surfaces of both structures. PZT patches provide the measurements of external excitation and dynamic responses. To derive two initial function matrices for the Euler–Bernoulli beam and WT blade with PZT patches, the same external loadings are applied to the structures, and their corresponding responses are obtained. Then, a total transfer function is achieved using two initial function matrices. At the final step, an AVC system is designed and applied to the governing dynamic equations of the beam with PZT patches to obtain the beam’s controlled lateral vibration domain under external loading and then to the WT blade with PZT patches using achieved transfer function.

### 3.1. The Euler–Bernoulli Beam Including Piezoelectric Patches

An assumed representative shape of the beam with patches is depicted in Figure 1.

The considered beam is a Euler–Bernoulli beam with length (*L*), width (*b*), and height (*h*), which is divided into n equal sections. To obtain the most realistic results, the boundary conditions of the beam are assumed as those of a cantilever beam, just like a blade in a WT, anchored at one end to a support. Edgewise vibrations are discarded in this study. Moreover, the PZT patches are assumed to be continuous along the transversal direction. The length of each PZT patch $L_{p}$ is as follows:(1)$$L_{p}=x_{2bj}-x_{1bj},$$
where *j* denotes the number of sensors; *b*, the beam; and 2 and 1, the beginning and the end of the patch, respectively. $x_{2bj}$ and $x_{1bj}$ denote the distance of the end tip and the beginning of the nth sensor–actuator from the base, respectively. In this study, the $L_{p}$ is taken as 10 cm for each sensor. Furthermore, the width of each PZT patch is the same as the width of the beam *b*, and it is located at the end of each section on the beam.

### 3.2. Analytical Modeling of the Euler–Bernoulli Beam with PZT Patches

The Euler–Bernoulli theory neglects shear deformations; hence, the shear tension and strain are excluded. In this theory, the shear force is only derived by this equation:(2)$$V=\frac{dM}{dx}.$$

The energy method and Lagrange equations are used to obtain the governing equation for the Euler–Bernoulli beam with PZT patches. Then, the assumed mode method is employed to solve the obtained equation. The total strain energy of the Euler–Bernoulli beam is as follows:(3)$$U=\frac{1}{2}\int_{0}^{L}\int_{A}^{}\left(\sigma_{xx}\varepsilon_{xx}\right)dAdx.$$

By inserting the associated equations for the stress and strain of the Euler–Bernoulli beam, Equation (Equation 3) can be represented as follows:(4)$$U=\frac{1}{2}\int_{0}^{L}D_{xx}{\left(\frac{\partial u_{z}^{2}\left(x,t\right)}{\partial x^{2}}\right)}^{2}dx,$$
where $D_{xx}$ is defined:(5)$$D_{xx}=EI_{x}.$$

$\sigma_{xx}$ and $\varepsilon_{x}$ are the normal stress and normal strain in the x-direction, respectively. E is the Young’s modulus, and $I_{x}$ is the second moment of area in the x direction. To derive the related strain energy equation for the beam, Equation (Equation 5) needs to be substituted into Equation (Equation 4) as follows:(6)$$U=\frac{1}{2}\int_{0}^{L}EI_{x}{\left(\frac{\partial u_{z}^{2}\left(x,t\right)}{\partial x^{2}}\right)}^{2}dx,$$
where $u_{z}\left(x\right)$ describes the transverse deflection of the beam at some position *x*. The total formulation of the Euler–Bernoulli beam’s kinetic energy is provided as follows:(7)$$T=\frac{1}{2}\int_{0}^{L}\int_{A}^{}\left(\rho \nu_{z}^{2}\right)dAdx,$$
where $\nu_{z}$ is velocity and it can be presented:(8)$$\nu_{z}=\frac{\partial u_{z}\left(x,t\right)}{\partial t}.$$

To formulate the Euler–Bernoulli beam’s cumulative kinetic energy and simplify it, Equation (Equation 8) could be replaced into Equation (Equation 7) as:(9)$$T=\frac{1}{2}\int_{0}^{L}\left(\rho A\left(x\right){\left(\frac{\partial u_{z}\left(x,t\right)}{\partial x}\right)}^{2}\right)dx.$$

By assuming the linear PZT constitutive relations and obtaining the governing equation of the vibration of the PZT patches, the stress and strain of these patches can be derived as follows:(10)$$\sigma_{xx}^{p}=c_{11}^{E}S_{1}-e_{31}E_{3},$$
(11)$$\varepsilon_{xx}^{p}=S_{1}=-y\frac{\partial^{2}u_{z}\left(x,t\right)}{\partial x^{2}},$$
where 1, 2, and 3 present the *X*, *Y*, and *Z* directions, respectively. $c_{11}^{E}$, $e_{31}$, and $E_{3}$ are modules of elasticity of the PZT in a constant electric field, PZT stress constant, and the electric field across the electrodes of the PZT, respectively. The following is the relation between the electrical field and the voltage applied to the PZT patch electrodes:(12)$$E_{3}=\frac{V(t)}{h_{a}},$$
where V(t) denotes the applied harmonic voltage. The correlation between the PZT stress constant and the associated strain constant, $d_{31}$, is provided as follows:(13)$$e_{31}=c_{11}^{E}d_{31}.$$

Thus, the equations of the strain energy and kinetic energy of a PZT patch are formulated as follows:(14)$$U_{j}^{p}=\frac{1}{2}\int_{x_{1a}}^{x_{2a}}\left(c_{11}^{E}I_{p}\right){\left(\frac{\partial^{2}u_{z}\left(x,t\right)}{\partial x^{2}}\right)}^{2}+J_{p}V{\left(\frac{\partial^{2}u_{z}\left(x,t\right)}{\partial x^{2}}\right)}^{2}dx,$$
(15)$$T_{j}^{p}=\frac{1}{2}\int_{x_{1a}}^{x_{2a}}\rho_{a}h_{a}b{\left(\frac{\partial u_{z}\left(x,t\right)}{2\partial x}\right)}^{2}dx.$$

The strain energy of the PZT patches must be added to the strain energy of the Euler–Bernoulli beam to achieve the overall strain of the Euler–Bernoulli beam along with the PZT sensor and actuator. The mentioned process must be applied for the kinetic energy of the PZT patches and the Euler–Bernoulli beam, respectively. The cumulative strain and kinetic energies of the beam with PZT patches can therefore be obtained as follows:(16)$$\begin{array}{c} U=\frac{1}{2}\int_{0}^{L}\left(EI_{x}{\left(\frac{\partial u_{z}^{2}\left(x,t\right)}{\partial x}\right)}^{2}\right)dx+\frac{1}{2}\int_{x_{1a}j}^{x_{2aj}}\left(c_{11}^{E}I_{p}\right){\left(\frac{\partial^{2}u_{z}\left(x,t\right)}{\partial x^{2}}\right)}^{2}+\\ J_{p}V{\left(\frac{\partial^{2}u_{z}\left(x,t\right)}{\partial x^{2}}\right)}^{2}{dx}_{j=1:n}, \end{array}$$
(17)$$T=\frac{1}{2}\int_{0}^{L}\left(\rho A\left(x\right){\left(\frac{\partial u_{z}\left(x,t\right)}{\partial x}\right)}^{2}\right)dx+\frac{1}{2}\int_{x_{1a}j}^{x_{2a}j}\rho_{a}h_{a}b{\left(\frac{\partial u_{z}\left(x,t\right)}{2\partial x}\right)}^{2}{dx}_{j=1:n},$$
where *j* indicates the number of piezoelectric patches.

#### Assumed Mode Method

The assumed mode approximation is used to solve the governing equations of the beam and the PZT patch. Under the actuation of a surface-bonded PZT patch, the transverse deflection can be expressed as follows:(18)$$\partial u_{z}\left(x,t\right)=\sum_{k=1}^{b}\Pi_{k}\left(x\right)q_{k}\left(t\right),$$
where $\Pi_{k}$ denotes the admissible function satisfying geometrical boundaries, and $q_{k}$ describes the corresponding unknown. By replacing Equation (Equation 18) with Equations (16) and (17), these
(19)$$\begin{array}{c} U=\frac{1}{2}\int_{0}^{L}EI\sum_{k=1}^{b}\sum_{l=1}^{b}\left(q_{k}{\Pi^{\prime \prime }}_{k}\right)\left(q_{l}{\Pi^{\prime \prime }}_{l}\right)+\\ {\left(\frac{1}{2}\int_{x_{1a}.j}^{x_{2a.j}}\left(c_{11}^{E}I_{p}\right)\sum_{k=1}^{b}\sum_{l=1}^{b}q_{k}q_{l}{\Pi^{\prime \prime }}_{k}{\Pi^{\prime \prime }}_{l}dx+\frac{1}{2}\int_{x_{1a}.j}^{x_{2a.j}}J_{p}V\left(t\right)\sum_{k=1}^{b}q_{k}{\Pi^{\prime \prime }}_{k}dx\right)}_{j=1:n}, \end{array}$$
(20)$$T=\frac{1}{2}\int_{0}^{L}\rho A\sum_{k=1}^{b}\sum_{l=1}^{b}\left({\dot{q}}_{k}\Pi_{k}\right)\left({\dot{q}}_{l}\Pi_{l}\right)+\frac{1}{2}\int_{x_{1a}.j}^{x_{2a}.j}\rho_{a}h_{a}b\sum_{k=1}^{b}\sum_{l=1}^{b}\left({\dot{q}}_{k}\Pi_{k}\right){\left({\dot{q}}_{l}\Pi_{l}\right)}_{j=1:n}.$$

The Lagrange equation is presented as:(21)$$\frac{d}{dt}\left(\frac{\partial T}{\partial {\dot{q}}_{j}}\right)-\left(\frac{\partial T}{\partial q_{j}}\right)+\left(\frac{\partial U}{\partial q_{j}}\right)=0.$$

The simple matrix form of the governing equation of the Euler–Bernoulli beam with the PZT patch is derived by substituting Equations (19) and (20) into the Lagrange Equation (Equation 21) as:(22)$$M\left\{\ddot{q}\right\}+K\left\{q\right\}=-V(t)\eta ,$$
where *M*, *K*, and η are presented as:(23)$$M=\frac{1}{2}\int_{0}^{L}\rho A\Pi_{k}\Pi_{l}dx+{\left(\frac{1}{2}\int_{x_{1a}.j}^{x_{2a.j}}\rho_{a}h_{a}b\Pi_{k}\Pi_{l}dx\right)}_{j=1:n},$$
(24)$$K=\frac{1}{2}\int_{0}^{L}\rho A{\Pi^{\prime \prime }}_{k}{\Pi^{\prime \prime }}_{l}dx+{\left(\frac{1}{2}\int_{x_{1a}j}^{x_{2aj}}\left(c_{11}^{E}I_{p}\right){\Pi^{\prime \prime }}_{k}{\Pi^{\prime \prime }}_{l}dx\right)}_{j=1:n},$$
(25)$$\eta ={\left(\frac{1}{2}\int_{x_{1a}j}^{x_{2aj}}J_{p}{\Pi^{\prime \prime }}_{i}dx\right)}_{j=1:n}.$$

### 3.3. Finite Element Modeling

The FE modeling of the GE 1.5 MW WT blade with PZT patches is constructed in an ANSYS code. The length of the blade is 45 m, which includes PZT patches. The PZT patches are set on the blade as in the Euler–Bernoulli beam, and the scheme is presented in Figure 2. The type of mesh used is hexahedral shell element (SHELL181) with four nodes and the size is 10 cm. For linear and/or large strain nonlinear applications, this mesh type is appropriate [73]. The connection between the PZT patches and the beam is assumed to be perfect. The blade with the same geometry of the beam used in the analytical modeling is modeled, and the simulation is conducted in a three-dimensional manner. In the combined field of transient structural dynamics and the electric field, the simulation of the model is performed. Defining the blade’s structural damping in FE models significantly influences the exact approximation of the real blade’s behavior. The Rayleigh damping is considered the structural damping model of the blade. The coefficients are obtained as α=0.0027 and β=0.0000017, which are the mass corresponding coefficient and stiffness corresponding coefficient, respectively.

### 3.4. The Wind Turbine Blade Projection on the Euler–Bernoulli Beam

Due to the fact that the velocity of wave propagation in solid media is only related to the frequency variants of actuation, not the time when forces are applied to a structure, we can consider both systems (the WT blade and the Euler–Bernoulli beam) as time-invariant. As both systems are linear time-invariant (LTI), the linear properties can be used to identify vibrational characteristics subjected to any external force. The LTI feature of the system is used to represent an efficient, feasible, and accurate model as a result of finding a semi-analytical solution for the wind turbine blade for deriving lateral blade deflections under various external forces. The proposed method leverages the LTI characteristics. Hence, only the LTI systems may use the proposed transfer projection function. The transmitted pulse signal is considered an impulse function to obtain the frequency responses of both the FE model of the blade and the analytical model of the Euler–Bernoulli beam. For the Euler–Bernoulli beam with PZT patches, a pulse signal is applied to every actuator’s location. Then, the n outcomes of pulse signals are obtained using the analytical equations of the beam by applying each pulse signal to an actuator. Instead of sending the transmitted pulse signal, the unit step function can be used in the LTI systems to identify structural frequency responses. Here, to solve the equations in the time domain, the unit step function is used. To obtain the system’s responses to the transmitted pulse signal, the achieved amplitude responses from n points are differentiated for each actuation. The frequency domain is considered to extract exact responses because many noises are present among the reached responses. Noises show small frequencies and are recognizable in the frequency domain. In addition, high frequencies caused by the resonance phenomenon can be eliminated in the frequency domain. To increase the ratio of signals to noises, overvoltage is required. A step function output is defined as $\partial y_{u}\left(t\right)$. To derive the responses to the impulse function, the outcomes need to be differentiated with respect to time. The response of the impact function $\partial y_{\delta }\left(t\right)$ is as follows:(26)$$\partial y_{\delta }\left(t\right)=\left(\frac{dy_{u}\left(t\right)}{dt}\right).$$

Then, the frequency response is as follows:(27)$$Y_{\delta }\left(\omega \right)=\mathrm{FFT}\left(\partial y_{\delta }\left(t\right)\right).$$

By applying the unit step function to each PZT actuator, n outputs from each of n sensors are obtained in every actuation (*n* is 45 in this study). Each element of the function matrix of the beam and blade is achieved as follows:(28)$$g_{ij}=\frac{{\mathrm{FFT}}_{\left(j_{th}\;output\right)}}{{\mathrm{FFT}}_{\left(i_{th}\;input\right)}},$$
where *j* denotes the number of sensor (output), and *i* indicates the number of actuator (input). Finally, the [N×N] matrix of function *G* is derived by repeating this procedure for every section
(29)$$G(i,j)=g_{i,j}\;\;\;\;\;i,j=1,2,\ldots ,n,$$
where $G\in R^{n\times n}$. Therefore, an [n×n] initial function matrix is derived for the Euler–Bernoulli beam with PZT patches using these external excitations and related responses. Similarly, an [n×n] initial function matrix is derived for the WT blade with patches by applying the same external excitations and deriving lateral movement. To obtain amplitude responses of the blade, the FE model of the WT blade with patches is constructed in an ANSYS code.

By performing the mentioned method on the beam and blade, two initial function matrices are achieved, $G_{b}$ and $G_{w}$, for the beam with PZT patches and the blade with PZT patches, respectively. In this step, the total transfer function matrix is derived by applying any similar external dynamic load, $U_{s}$, to both systems, and the concerning linear feature of the whole system is as follows: (30)$$Y_{b}=G_{b}\cdot U_{s},$$
(31)$$Y_{w}=G_{w}\cdot U_{s},$$
by using linear feature of the whole system: (32)$$Y_{w}=\left(\frac{G_{w}}{G_{b}}\right)\cdot Y_{b}.$$

Next, the total transfer function matrix is obtained using two initial matrices that are achieved in the previous step as follows: (33)$$G_{T}=\left(\frac{G_{w}}{G_{b}}\right).$$

The total transfer function $G_{T}$ can project vibrational and tensional outcomes of the analytical solution of the Euler–Bernoulli beam with PZT patches to the WT blade with PZT patches under the same external loading (Equations (31)–(33)). The proposed method can be organized as a semi-analytical solution to derive lateral deflection of the blade to external forces (Figure 3). The steps for the determination of lateral movements of the WT blade using a semi-analytical method are as follows:Step 1. Deriving governing equations of the Euler–Bernoulli beam with the attached PZT actuator and sensor using the energy method.Step 2. The derived governing equations are solved using an assumed mode method.Step 3. The FE model of the WT blade with PZT patches is developed.Step 4. Obtaining frequency responses of both systems, the FE model of the blade, and the Euler–Bernoulli beam by applying the transmitted pulse signal.Step 5. Deriving initial function matrices for the beam and the blade.Step 6. Obtaining total transfer function using two achieved initial function matrices regarding the linear properties of the systems.

### 3.5. The Implementation of Control System

The LQR is considered in this study to damp unwanted lateral vibration of the desired structure. The controlling rules of the control system are used for the WT blade using PZT patches. The LQR is a common design technique that provides practical feedback gains. The LQR is one of the optimal control techniques that consider the states of the dynamic system and control input to make the optimal control decisions. This is both simple and robust [74,75]. The purpose of the LQR controlling method is to determine a state-feedback optimal control force, which can minimize the definite quadratic cost function (Equation (Equation 35)). The weighted matrices of the control performance index used to design optimal state-feedback gains are usually determined by trial and error via simulation. Authors mainly randomly select *Q* and *R* to determine whether they meet simulation requirements. The purpose of this section is to obtain needed matrices in an LQR controller to design an optimal AVC system. To derive LQR, the generic form of the system can be written in a state–space form:(34)$$\dot{X}\left(t\right)=AX\left(t\right)+BU\left(t\right),$$
in which all of the *n* states X(t) are available for the controller. The optimal state-feedback controller or LQR minimizes the infinite quadratic cost function Equation (Equation 35), including the state variables (X(t)) and control action (U(t)). Here, [Q,R] are the symmetric positive semi-defined weighting matrix and the positive weighting factor, respectively, that tune the penalty on the excursion of state variables and control action. To minimize the integral performance index Equation (Equation 35), the continuous time algebraic Riccati equation Equation (Equation 36) can be used:(35)$$J=\int_{0}^{\;}\left[\left(X^{T}\left(t\right)QX\left(t\right)\right)+\left(U^{T}\left(t\right)RU\left(t\right)\right)\right]dt,$$
(36)$$A^{T}P+PA-PBR^{-1}B^{T}P+Q=0.$$

The state-feedback optimal control force U(t) is [76]:(37)U(t)=−KX(t),
where
(38)$$K=R^{-1}B^{T}P.$$

*K* is unknown as an optimal controller gain, and *P* denotes the unique, symmetric, positive semi-definite solution to the algebraic Riccati equation (Equation (Equation 36)). The term $X^{T}\left(t\right)QX\left(t\right)$ is a measure of control precision, and the term $U^{T}\left(t\right)RU\left(t\right)$ is a measure of control effort.

In order to implement the AVC system with LQR on the WT blade, apply controlling rules to the structure, and obtain controlled lateral vibrations, the proposed mapping method contains the Euler–Bernoulli beam is used. The governing equations of the Euler–Bernoulli beam with PZT patches were derived in this study (Equations (1)–(25)). The number of assumed modes in Equation (Equation 18) is set to be n=4, since the convergence for results was achieved with the sum of 4 number of modes. Then, X(t) is as follows:(39)X(t)=q(t)q˙(t)=q1(t)q2(t)q3(t)q4(t)q1˙(t)q2˙(t)q3˙(t)q4˙(t)

The vector differential equation of Equation (Equation 22) can be written as:(40)$$M\left(\begin{array}{c} \ddot{q_{1}}\left(t\right)\\ \ddot{q_{2}}\left(t\right)\\ \ddot{q_{3}}\left(t\right)\\ \ddot{q_{4}}\left(t\right) \end{array}\right)+K\left(\begin{array}{c} q_{1}\left(t\right)\\ q_{2}\left(t\right)\\ q_{3}\left(t\right)\\ q_{4}\left(t\right) \end{array}\right)=F_{act}+F_{ext},$$
where $F_{act}$ = −V(t)η and $F_{ext}$ is an external excitation loading. The system matrix, *A* and the input matrix, *B* of the state-space form of Equation (Equation 40) are as follows:(41)$$A=\left[\begin{array}{cc} \;0_{4\times 4} & I_{4\times 4}\\ \;{M^{-1}K}_{4\times 4} & 0_{4\times 4} \end{array}\right];B=\left[\begin{array}{c} \;0_{4\times 2}\\ \;{M^{-1}U}_{4\times 2} \end{array}\right],$$
where *M* and *K* denote the system mass and system stiffness, respectively, that *M* and *K* were derived in Equations (23) and (24).

The inputs of LQR are *A*, *B*, *Q*, and *R* matrices and the output is *K*. By changing *Q* and *R* in each effort, a new *K* is achieved for Equation (Equation 38). The new *K* is then substituted into Equation (Equation 37) and results in a new state-feedback optimal control force (U(t)). By substituting U(t) into Equation (Equation 34), the equations for the corresponding unknowns ($q_{k}$) are derived. The controlled flap-wise displacements of the Euler–Bernoulli beam with PZT patches in each nodal location for a certain period of time are obtained by substituting all $q_{k}$ into Equation (Equation 18) and solving the achieved equations. In order to apply LQR controlling rules to the WT blade and obtain controlled lateral movements, the proposed mapping method is employed. The total transfer function matrix $G_{T}$ (Equation (Equation 33)) can project the Euler–Bernoulli beam’s controlled lateral displacement to the WT blade when both systems are excited with the same excitation loading. Excitation loadings, including applied external forces and provided controlling forces by PZT actuators for both structures, will be the same during exciting structures and controlling phases. The admissible computational cost is the main advantage of using the proposed mapping method for applying controlling rules to the WT blade. Figure 4 presents the block diagram for the control loop simulation.

## 4. Results and Discussion

The Euler–Bernoulli beam and the FEM model of the WT blade were loaded separately with three external different loadings to validate the proposed semi-analytical solution using the transfer function matrix method for the WT blade. A sine distributed load with an amplitude of 20 $\mathrm{kgf}/m^{2}$, frequency of 2 Hz, and attack angles of 15°, 30°, and 45° were applied, respectively. The amplitude and frequency amounts of the excitation were selected via field testing data and condition monitoring [77]. The Reilly–Ritz method was used to discretize a continuous force generated by the applied external loadings to the number of divided pieces (number of actuators–sensors) on the Euler–Bernoulli beam. The comparisons of analytical and FEM results of the 20th and 35th sensors are presented in Figure 5 and Figure 6, respectively.

Figure 5 and Figure 6 present an acceptable compliance between analytical and FEM results. Furthermore, the error between the results of the analytical solution and FEM for each sensor by applying the mentioned loadings is presented in Figure 7, Figure 8 and Figure 9. In this study, RMSE is used to express the model error for each sensor. RMSE is a quadratic scoring rule and a standard method to calculate the error of a model. Because errors in RMSE are squared before they are averaged, RMSE gives higher weight to large errors. This function makes RMSE more useful when large errors are particularly undesirable.
(42)$$\mathrm{RMSE}=\sqrt{\sum_{i=1}^{N}\frac{{\left(y_{i}-{\hat{y}}_{i}\right)}^{2}}{n}},$$
where for each sensor, $y_{i}$ is *i*’ outcome value from FEM and ${\hat{y}}_{i}$ is *i*’ outcome value from the proposed method and n is number of observation.

According to the results, there are no precise patterns in the error rates depending on sensor locations (Figure 7, Figure 8 and Figure 9). The results indicated that as the sensors’ distance from the base of the WT increases, the method’s accuracy diminishes; nevertheless, there are certain outliers to the patterns (Figure 7, Figure 8 and Figure 9). The vibration domain of the structure naturally grows with the increase in the distance from support, which is one cause for obtaining escalating rate of errors by raising the sensors’ interval from the base. Therefore, the span from the base can be considered a determining factor for the method’s precision. In order to avoid neutralizing the sensing and actuating effects of the PZT patches, they are located at the same distance from each other in this study. Considering the current set up of the sensors on the structures, the performance of the mapping method is more reliable for the distance from sensor number 1 to 11 and from sensor number 16 to 33 (the error less than 15%). We also noticed, based on the patterns (Figure 7, Figure 8 and Figure 9), that the performance of the method decreases with a steep slope after sensor number 36. In addition, the results show a sensible issue for the method for the distance from sensor number 13 to 15. Changing the vibration domain along the structures, and local vibration nodes and peaks, may cause unexpected outliers in the pattern of the method’s performance.

The control system is implemented on the blade using the transfer function method and the Euler–Bernoulli beam in this study, as using the FE modeling directly to apply control rules on complex shape structures is tremendously time-consuming. An LQR was designed as a proper control system because the considered structural system in this study is time-invariant. To obtain suppressed transverse deflection of the WT blade following the application of the designed LQR rules, the proposed mapping system, including the Euler–Bernoulli beam and the transfer function method, was used. In the design process of the LQR control method, the *Q* and *R* matrices were assumed as unity matrix and γI, respectively, where Gamma is a scalar. In this study, we continue to adjust R until it meets the requirements, which includes damping flap-wise deflection and using minimum energy as needed voltage for PZT patches as much as possible. The outcomes of damping vibration of the 15th, 30th, and 45th sensors on the blade under loading number 2 are provided as follows:

The task of the proposed AVC system, containing an LQR control method and PZT actuator, is to suppress the vibration magnitude of the nodes. The vibration amplitudes of the uncontrolled and controlled models are plotted for different values of *R* with *Q* = *I* for sensors number 15, 30, and 45 in Figure 10, Figure 11 and Figure 12, respectively. Effect of selection of weighting matrix R of LQR on the vibration domain of sensors 15th, 30th and 45th on the blade are presented in Figure 13, Figure 14 and Figure 15 for two different Gamma amounts ($\gamma_{1}$ = $10^{-11}$ and $\gamma_{2}$ = $10^{-13}$). The results indicate that selection of the weighting matrix *R* plays a significant role in designing an optimal controller. It has been observed that the amplitudes of LQR controller have considerably died down, while the weights on the control input in the cost function decreased. When matrix *R* is diminished, the control energy spent increases, but the settling time decreases. The proposed AVC system performs satisfactory for alleviating vibrations along the WT blade even at the tip of the blade, which has the maximum amount of lateral vibration amplitude.

## 5. Conclusions

In this study, a smart AVC system that included an LQR control and the PZT actuator was proposed to damp the transverse deflection of a WT blade. To implement active control rules on the WT blade, an effective semi-analytical solution based on a unique transfer function matrix was developed to undertake the projection of the WT blade to a Euler–Bernoulli. The energy method was employed to derive the governing equations of the Euler–Bernoulli beam with the attached PZT actuators and sensors. The obtained governing equations were then solved using the assumed mode method. Rayleigh damping factors were used to develop the FE model of the WT blade with PZT patches. For the linear feature of the whole system, the final transfer function matrix was derived by applying the same external dynamic force to both systems. In order to validate the mapping performance of the transfer function matrix, the Euler–Bernoulli beam and the FE model of a WT blade were separately loaded with three different loads. The results indicate satisfactory correlations between the deflections directly derived from the FE model of the WT blade and the movements achieved from the proposed mapping method. To damp the lateral vibration of the WT blade, the proposed AVC system was used. The study demonstrated satisfactory performance of the proposed AVC on damping vibrations of the WT blade. Hence, in addition to WT blades, this article may be helpful for investigating more complex dynamic systems for their AVC, right from their modeling to optimal controller design. Further studies will consider an optimal number of PZT actuators and sensors with optimization of their placements, as well as optimal static output feedback controllers. Finally, a prototype of the proposed AVC and the mapping method for a WT blade will be developed, and the experimental investigation will be carried out to verify the AVC system performance of the prototype. 

## Figures and Tables

**Figure 1 sensors-22-05691-f001:**
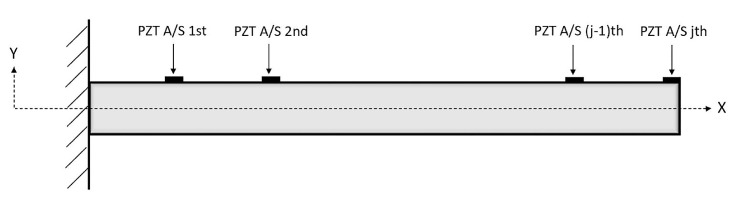
Representative shape of the beam including piezoelectric patches.

**Figure 2 sensors-22-05691-f002:**
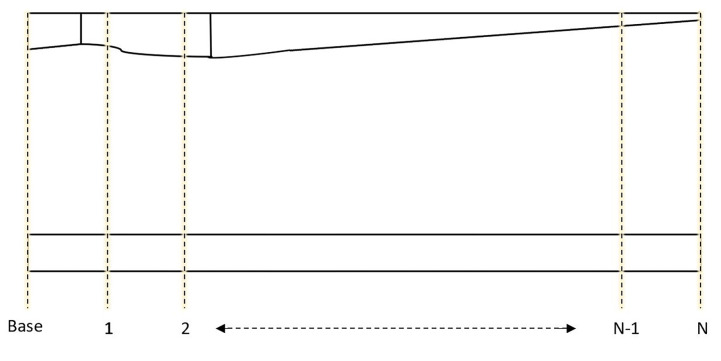
Section’s estimated diagram from the blade to the beam.

**Figure 3 sensors-22-05691-f003:**
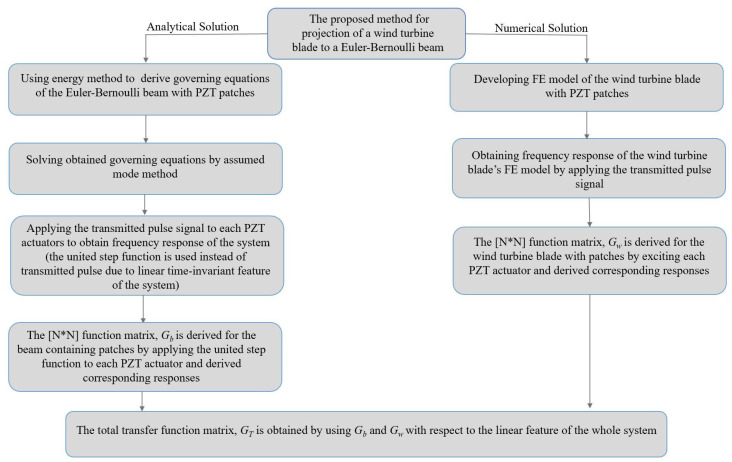
Flow chart of the total transfer function deriving process.

**Figure 4 sensors-22-05691-f004:**
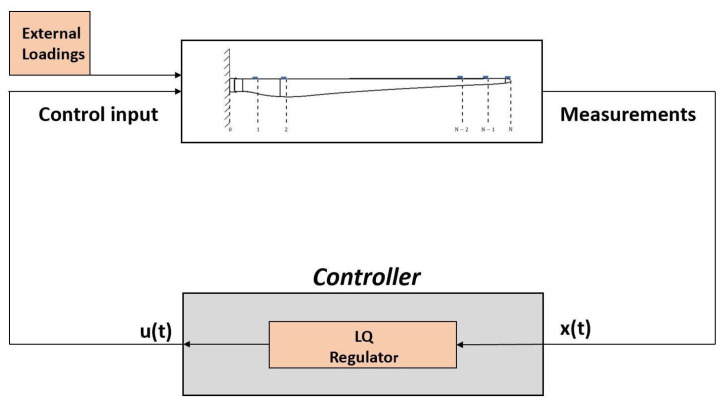
Block diagram for control.

**Figure 5 sensors-22-05691-f005:**
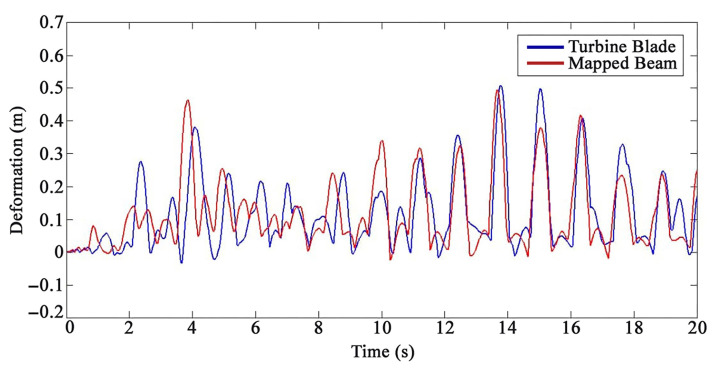
The outcomes of the analytical results and FEM results for the twentieth sensor under loading number 1 (15 degrees attack angle).

**Figure 6 sensors-22-05691-f006:**
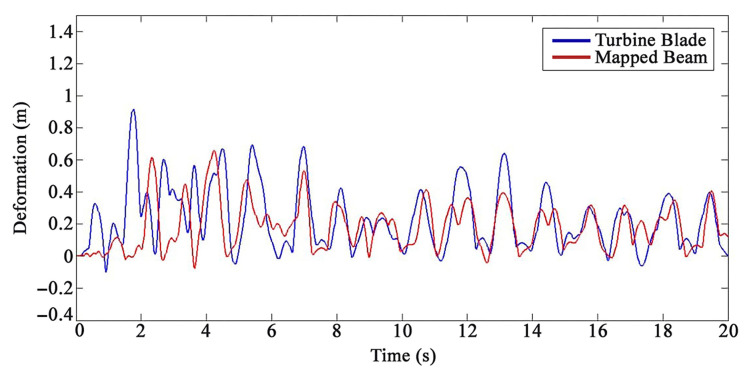
The outcomes of the analytical results and FEM results for the thirty-fifth sensor under loading number 2 (30 degrees attack angle).

**Figure 7 sensors-22-05691-f007:**
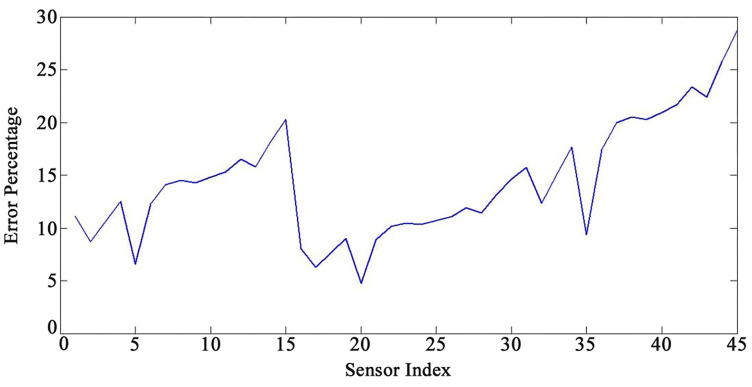
Error percentage for each sensor under loading number 1 (15-degree attack angle).

**Figure 8 sensors-22-05691-f008:**
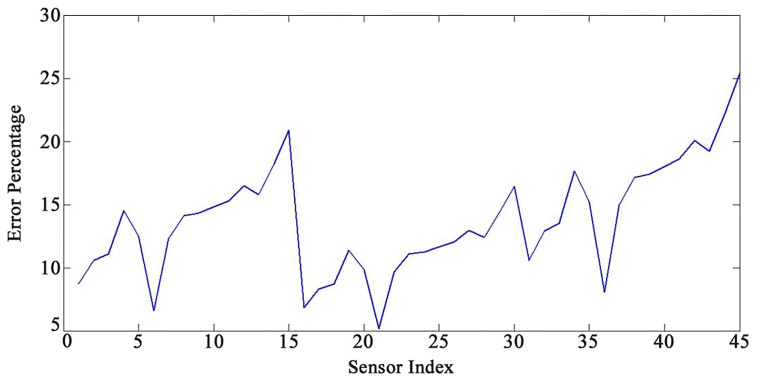
Error percentage for each sensor under loading number 2 (30-degree attack angle).

**Figure 9 sensors-22-05691-f009:**
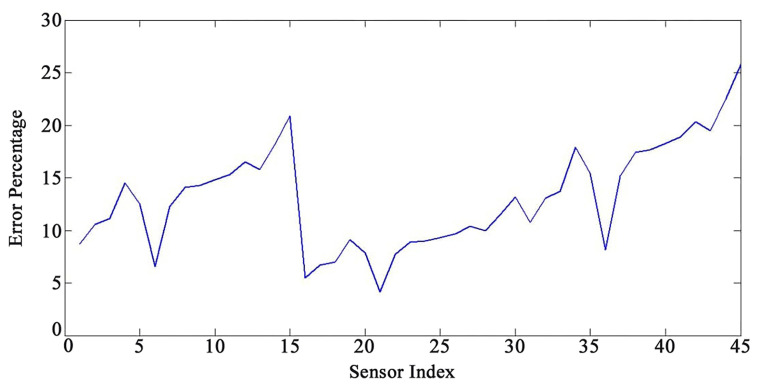
Error percentage for each sensor under loading number 3 (45-degree attack angle).

**Figure 10 sensors-22-05691-f010:**
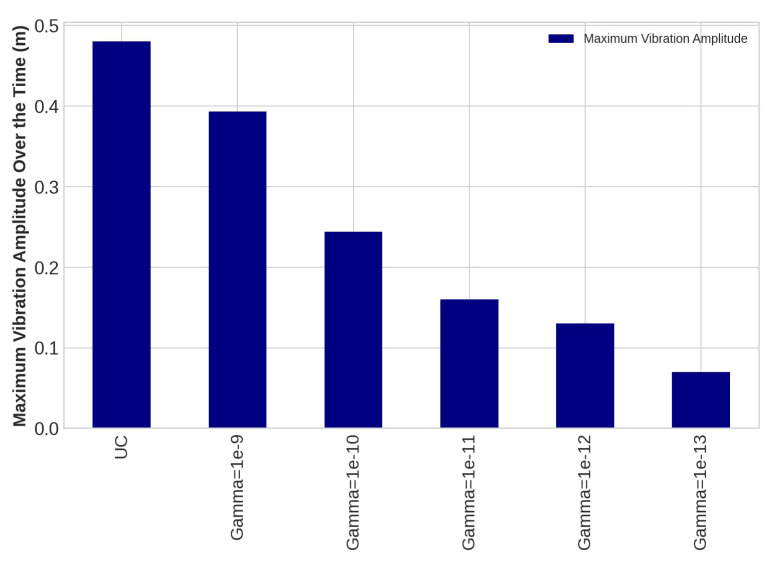
The outcomes of applying controlling rule for the fifteenth sensor under loading number 2 (30-degree attack angle).

**Figure 11 sensors-22-05691-f011:**
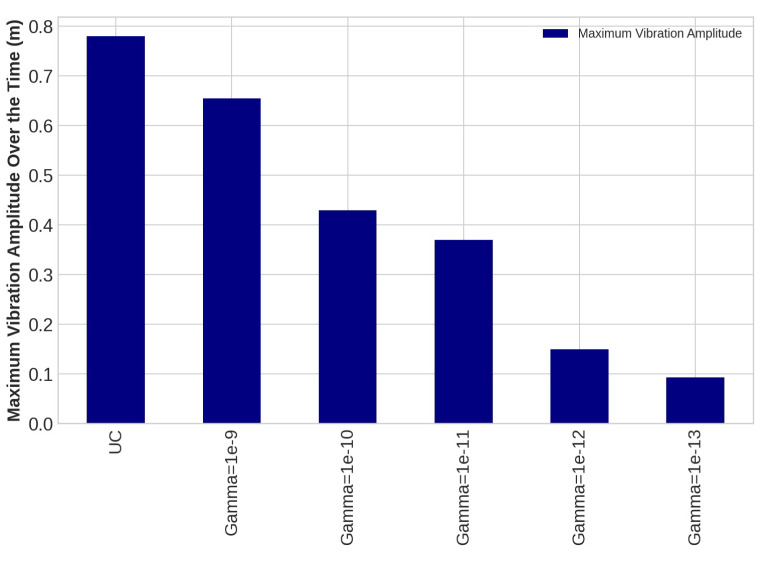
The outcomes of applying controlling rule for the thirtieth sensor under loading number 2 (30-degree attack angle).

**Figure 12 sensors-22-05691-f012:**
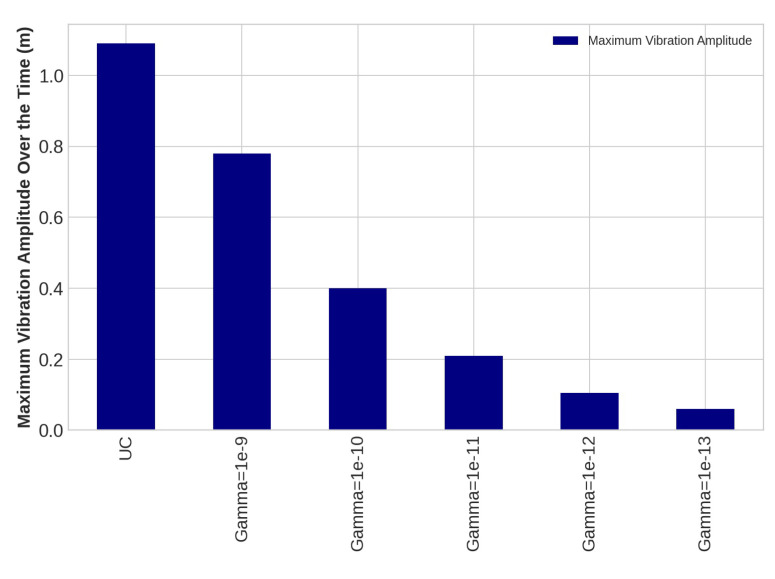
The outcomes of applying controlling rule for the forty-fifth sensor under loading number 2 (30-degree attack angle).

**Figure 13 sensors-22-05691-f013:**
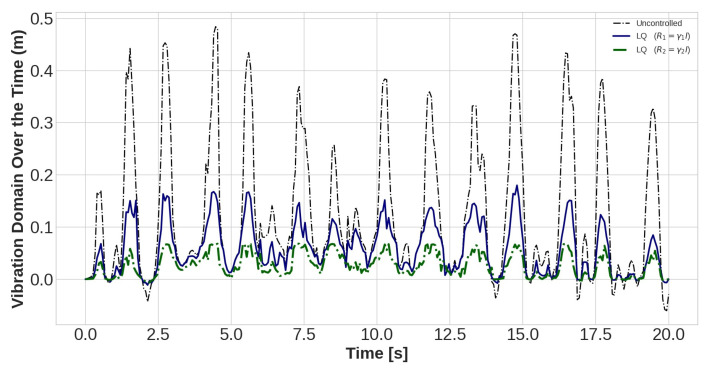
The lateral vibration domain with 2 different Gamma amounts ($\gamma_{1}$ = $10^{-11}$ and $\gamma_{2}$ = $10^{-13}$) for the fifteenth sensor under loading number 2 (30-degree attack angle).

**Figure 14 sensors-22-05691-f014:**
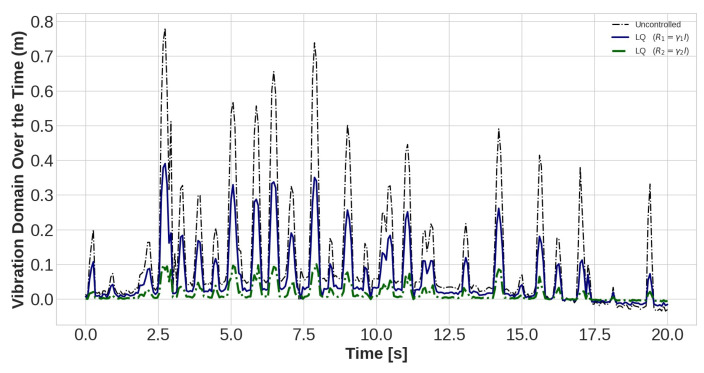
The lateral vibration domain with 2 different Gamma amounts ($\gamma_{1}$ = $10^{-11}$ and $\gamma_{2}$ = $10^{-13}$) for the thirtieth sensor under loading number 2 (30-degree attack angle).

**Figure 15 sensors-22-05691-f015:**
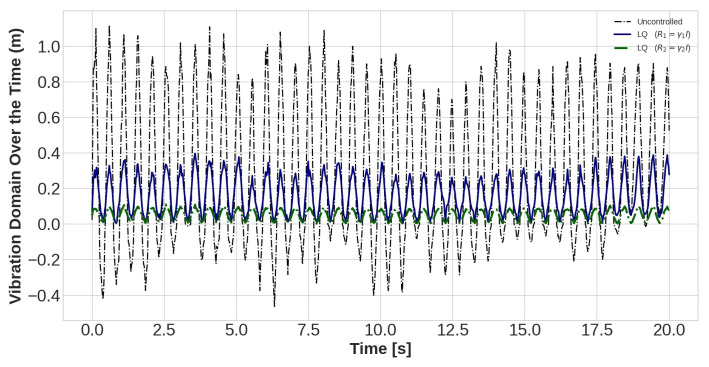
The lateral vibration domain with 2 different Gamma amounts ($\gamma_{1}$ = $10^{-11}$ and $\gamma_{2}$ = $10^{-13}$) for the forty-fifth sensor under loading number 2 (30-degree attack angle).

## Data Availability

Not applicable.

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
