# Peer review of "Smart Active Vibration Control System of a Rotary Structure Using Piezoelectric Materials"

_sensors, 2022, doi:10.3390/s22155691_

Round 1

Reviewer 1 Report

Controlling the vibrations in rotary structures is a challenging work. so the author's work is meaningful for this field. This paper proposed a smart AVC system that included an LQR control and the PZT actuator to control the transverse deflection of a WT blade. However a few improvements are needed. The following revisions are recommended:

(1) A concise and factual abstract is required. The abstract should state briefly the purpose of the research, the principal results and major conclusions.

(2) It is suggested to explain the error between the results of the analytical solution and FEM in Figures 7-9.

(3) The obtained results indicate that the proposed smart control system can efficiently suppresses lateral vibrations. It is suggested to conduct the experiment to verify the vibration control system effectiveness.

(4) The conclusion is too long to be shortened to specific points. Moreover, try to suggest some possible extensions to the work. This is very important and needs to be included in conclusions as it sets the correct premise for future directions.

(5) There are some grammar and spelling mistakes inside the manuscripts. The authors are suggested to make a careful check again.

Reviewer 2 Report

The paper is devoted to the presentation of the method dedicated for more efficient analysis of the dynamic properties of wind turbine blade making use of the Euler–Bernoulli beam theory and declaration of a transfer function matrix. The Authors propose a semi-analytical method applying analytical solution of vibration analysis adapting the model of a piezoelectric transducer acting as actuators and sensors. The method makes use the referential results obtained using FE code. Finally, the developed modelling approach is employed to study the case of reduction of transverse vibration of a given wind turbine blade model.

The presented method is interesting, and the work, in the reviewer’s opinion, is worth to be published. A practical engineering problem is identified – reduction of vibrations in wind turbines. In fact, the Authors focused on the feasible implementation of the developed algorithms for active control of vibrations. However, the below-stated issues should be considered before publication. Moreover, minor spell check is required.

The first section “Introduction” suffers from lack of citations.

Row 26: There are no citations provided at the end of the sentence “Many studies have used passive, …” to support it.

35: similar comment for the sentence “PZT transducers have been widely used …”

38: what does it mean “perfect layers”?

The discussion on the FE model (as providing referential results) mesh quality should be addressed in the manuscript.

Although the results may be considered as satisfactory, when comparing the FEA and the outcomes from the proposed approach, more comprehensive discussion on the sensors (virtual sensors) should be carried out to more reliably use the method. The issue should be commented in the paper.

Flaws, minor issues:

Abstract, row 17: I am confused with the statement “The obtained results indicate that the proposed smart control system contains PZT patches,…”. It was already mentioned that the proposed solution assumes the use of PZT ceramic patches, so what is a new information presented here?

23: “vibration” -> “vibrations”

37: “… systems owning to their special.” There is missing something the statement.

62: There is something wrong about the expression “gained more popularity owing to” in the sentence  “Bigger, complex WTs have often gained more popularity owing to …”

79: “blade” ->”blades”

102: “normal frequencies” -> “natural frequencies”

109: ‘support”->” supported”

119: “body.By”->” body. By”

214: “plane sections remain plane” -> “plane sections remain parallel” or similar and without quotation signs

246: “3.1. The Euler-Bernoulli Beam Includes Piezoelectric Patches” -> “3.1. The Euler-Bernoulli Beam Including Piezoelectric Patches” or similar

247: “An anticipated shape”->” An assumed representative shape”

Caption Figure 1: “Representative shape of the beam including piezoelectric patches” or similar

248: “The beam being considered”->” The considered beam”

249: “n-equal”->” n equal”

249: “best“->”most realistic”

256/257: “10 cm” should not be split at the line break

260: “shearing deformations”->” shear deformations”

260: “shearing tension”->” shear tension”

270: “Young module”->” Young’s modulus”

331: “n-outcomes”->” n outcomes”

336: “n-points”->” n points”

344: “n-outputs from each n-sensors”->” n outputs from each of n sensors”

Eq. 34: why is the X (with prime) at the left side of the equation capital?

Figure 10: what does it mean “vibration domain”? – amplitude?

Reviewer 3 Report

In this paper, the authors present a numerical study on the vibration suppression of wind turbine blades applying active vibration control via pairs of PZT sensors-actuators. An analytical solution based on Euler Bernoulli beam theory is compared against FEM results and the control scheme is numerically tested. The paper is well written without language errors. However, the introduction needs some updating and the results should be better presented. Please see major and minor comments for revision:

Major Comments

the introduction should contain more recent reviews on vibration control of wind energy systems, e.g. Malliotakis, G.; Alevras, P.; Baniotopoulos, C. Recent Advances in Vibration Control Methods for Wind Turbine Towers. Energies 202114, 7536. https://doi.org/10.3390/en14227536

I think that the results section should focus more on the achieved vibration suppression rather than the corroboration between FEM and the analytical solution. Besides, its not clear how this is related to the control scheme. I suggest to add results from more nodes showing the reduction of vibration with the control. The nodes should be carefully selected and justified based on practical problems of the structure.

Its not clear how the function G_T in Eq. 33 is actually used in the control scheme described in section 3.6

Minor comments

Lines 415-416: the excitation amplitude is given as 20 kg/m3 which corresponds to mass. Please clarify.

Figure 3: In the analytical solution flowchart there is a typo: is derived for the beam containing patches.

Figure 11 should be produced with higher quality.

Round 2

Reviewer 1 Report

This paper have been revised according to the referees’ comments. It can be accepted.

Reviewer 3 Report

the authors have performed extensive revisions to satisfactorily address my comments from the previous review round. I recommend the paper for publication as is.